# Data augmentation via warping transforms for modeling natural variability in the corneal endothelium enhances semi-supervised segmentation

**Sergio Sanchez**[1], **Noelia Vallez**[2], **Gloria Bueno**[2], **Andres G. Marrugo**[1]*

**1** Facultad de Ingeniería, Universidad Tecnologica de Bolivar, Cartagena, Colombia, **2** VISILAB, Universidad de Castilla-La Mancha, E.T.S. Ingeniería Industrial, Avda Camilo Jose Cela, Ciudad Real, Spain

* agmarrugo@utb.edu.co

**Data Availability Statement:** The data used in this paper has been sourced from the project "New classification of specular microscopy images of the corneal endothelium in Fuchs dystrophy through

## Abstract

Image segmentation of the corneal endothelium with deep convolutional neural networks (CNN) is challenging due to the scarcity of expert-annotated data. This work proposes a data augmentation technique via warping to enhance the performance of semi-supervised training of CNNs for accurate segmentation. We use a unique augmentation process for images and masks involving keypoint extraction, Delaunay triangulation, local affine transformations, and mask refinement. This approach accurately captures the natural variability of the corneal endothelium, enriching the dataset with realistic and diverse images. The proposed method achieved an increase in the mean intersection over union (mIoU) and Dice coefficient (DC) metrics of 17.2% and 4.8% respectively, for the segmentation task in corneal endothelial images on multiple CNN architectures. Our data augmentation strategy successfully models the natural variability in corneal endothelial images, thereby enhancing the performance and generalization capabilities of semi-supervised CNNs in medical image cell segmentation tasks.

## Introduction

The segmentation of corneal endothelial cell images is important for assessing corneal health and diagnosing various corneal diseases based on cell morphology [1]. Accurate segmentation is still challenging despite the development of automated methods over the last decade [2]. Recent automated methods based on Deep Learning [3, 4] have superseded traditional methods [5] based on morphological operations, contour detection, and spatial frequency analysis due to their improved performance. However, their success and generalization capabilities largely depend on expensive image annotation by experts due to their supervised nature.

In this context, the segmentation of microscopic images of the corneal endothelium, as illustrated in Fig 1, faces a series of challenges. These include data acquisition issues such as calibration problems, noise, and equipment handling errors [6, 7]. Image processing is further complicated by factors like variations in lighting, shadows, blurring, glare, and unwanted

artificial intelligence techniques". The anonymized dataset, ensuring privacy and confidentiality, is available at the Open Science Framework repository (https://osf.io/75kmu/), DOI: 10.17605/OSF.IO/75KMU.

**Funding:** Dr. Andres G. Marrugo and Mr. Sergio Sanchez received funding from the Departamento Administrativo de Ciencia, Tecnología e Innovación (COLCIENCIAS), under grant number 763-2021. Dr. Noelia Vallez and Dr. Gloria Bueno received funding from the European Union NextGenerationEU/PRTR, under grant reference 2022-GRIN-34352. The funders had no role in study design, data collection and analysis, decision to publish, or preparation of the manuscript.

**Competing interests:** The authors have declared that no competing interests exist.

artifacts [8]. At the medical evaluation stage, specialists grapple with issues like misdiagnoses and discrepancies between experts, compounded by subjectivity in image interpretation and lack of accurate annotations [9]. The complexity is increased in images of endothelial cells affected by diseases such as Fuchs dystrophy, presenting additional diagnostic challenges [10].

Due to the complex nature of medical imaging and its high cost of labeling, there has been significant growth in research into data augmentation techniques that seek to improve the performance of CNN in the medical image segmentation task [11–13]. These techniques aim to enrich the diversity and representativeness of the training data set, which in turn contribute to better generalization and accuracy of the CNN. However, not all data improvement techniques are effective. To address this limitation, more advanced approaches based on semi-supervised learning and data augmentation techniques are being explored and have proven to be a promising solution. These networks have the ability to extract relevant features from a large set of unlabeled images, making it possible to capture fundamental patterns and structures present in medical images. Which they subsequently use to perform fine tuning with a limited number of labeled images to learn the [14] segmentation task.

Hence the importance of using data augmentation strategy that allows applying transformations that manage to simulate the natural variations present in the images, that help to enrich the data sets and improve the robustness of the segmentation models. These transformations must not only be realistic, but also specific to the task and anatomy at hand. For example, in the case of segmentation of the corneal endothelium, it is critical that the applied deformations reflect actual variations in cell morphology and surrounding tissue characteristics.

Therefore, in this article we present a novel and effective methodology that addresses the inherent challenges of low-annotation medical databases, with cutting edge results. We propose a data augmentation technique using semi-supervised learning to improve the segmentation task in specular microscopy images of the corneal endothelium. Our main contribution focuses on the implementation of the warping and watershed strategy to model and generate transformed images and masks similar to the original, contributing to the challenge of sparsely

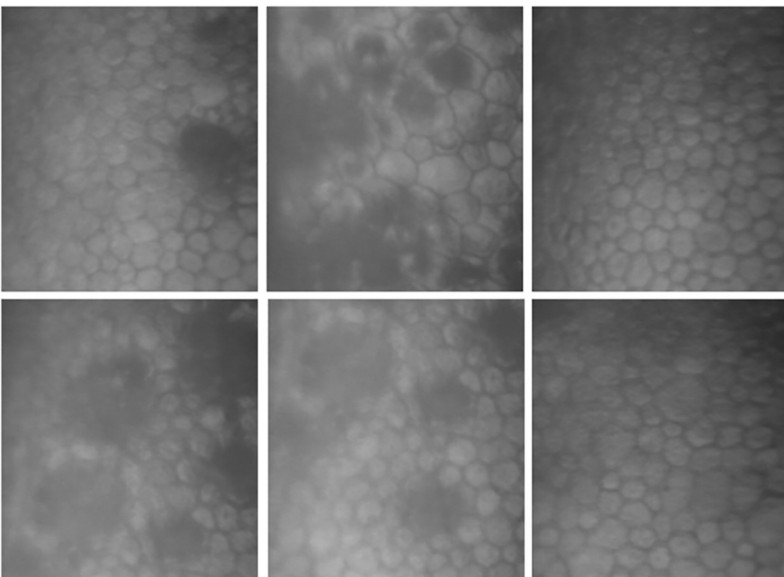

**Fig 1. Illustration of various challenges in corneal endothelium imaging including low visibility, blurring, and images with guttae due to Fuch's dystrophy, highlighting the complexities in data acquisition and analysis.**

annotated medical databases. In the following sections, the proposed method will be described in detail and a comparison with traditional strategies will be made.

## Related work

Image segmentation of the corneal endothelium is a problem that has been the subject of research for several decades. This task faces multiple challenges, as images of these cells often present blur, low contrast, and irregular lighting. Over time, various techniques have been developed with the aim of achieving a more precise delineation and effective separation of endothelial cells. The first strategies were based on manual segmentation, which is prone to human error [15, 16]. Subsequently, techniques based on thresholding methods, filters and image processing were used. Although they improved segmentation, there were still challenges with low-quality images and noise [17, 18]. However, with advances in the field of computer vision and deep learning, there has been a significant change in the way segmentation of the corneal endothelium is approached.

Convolutional Neural Networks have demonstrated an exceptional ability to automatically learn intrinsic features in this type of data. In recent years, different authors such as Okumura et al., and Sierra et al., [3, 19] have investigated deep networks based on supervised learning, achieving plausible results, however, these models require a large volume of images annotated by specialists to improve performance, they are sensitive to domain changes and overfitting. Not to mention that these databases are often sparse, unbalanced, require informed consent from patients, and are complex to acquire and interpret [20].

To overcome the problem of unlabeled medical databases, there are different strategies, such as regularization methods, transfer learning, one-time and zero learning algorithms, semi-supervised learning and data augmentation [21, 22].

Regularization methods are commonly used in deep network configuration, some strategies such as dropout, early stopping and batch normalization control the complexity of the neural network, avoid overfitting and avoid excessive dependence between neurons. However, its effectiveness may be decreased by the size of the data set, network architecture, learning rate, optimization algorithm, training iterations, and hyperparameter settings [23, 24].

Transfer learning takes advantage of the knowledge of frozen deep neural networks, which have been trained with millions of non-medical images. However, this approach may not provide features directly relevant to medical segmentation tasks. Despite this, transfer learning can be advantageous, as previously trained networks have learned low-level visual features such as borders, textures, and basic patterns, useful for further tuning [25, 26]. Nonetheless, this strategy presents challenges, mainly to ensure the relevance and applicability of the pre-trained features in the specific context of medical imaging.

Few-shot learning algorithms, including one-shot and zero-shot learning, are machine learning strategies that offer promising solutions to the scarcity of unlabeled medical databases. These strategies have achieved promising results in the performance of deep neural networks [27, 28]. These techniques, despite their innovative use of minimal data and ancillary information, are often hampered by bias and lack of diversity in the training data.

Semi-supervised learning is a strategy that combines labeled data with unlabeled data during the training of a neural network. Initially it learns unlabeled data features, then the weights are frozen and finally used in a fine-tuning stage to learn a specific task [29, 30]. These architectures are used on generative approaches [31], predictive tasks [32], contrastive and non-contrastive learning [33] and bootstrap approaches [34, 35]. Most of these techniques require heuristics and fine tuning of hyperparameters, so each approach may perform differently depending on the type of problem being addressed. Which is why it is important to obtain

more labeled data, either through conventional or synthetic means, to ensure better performance of CNNs.

Data augmentation is a strategy to increase the volume, quality, and diversity of annotated images, though its application varies depending on the task. This technique is commonly used with both traditional methods and deep learning. Traditional approaches typically involve geometric transformations (e.g., rotation, flipping, cropping, shifting, zooming, and random local rotation) [36], photometric adjustments (color space shifting, brightness) [37, 38], and noise injection or filtering [39, 40]. Yet, these methods often fail to fully capture the natural variability of biological structures, limiting their effectiveness in more complex segmentation tasks.

Neural network-based data augmentation, using architectures like GANs and style transfer [41, 42], enables the generation of data that is often indistinguishable from real data, addressing challenges in image processing. These networks have been widely adopted to increase data volume; however, they require substantial computational resources and often produce artifacts or unnatural shapes in the output [43–45].

Several data augmentation strategies for segmenting corneal endothelium images from specular microscopy have been explored in the literature. Authors like Sierra et al. [46], Vigueras et al. [47], Kolluru et al. [48], and Shilpashree et al. [49] have used basic geometric transformations such as rotations, translations, cropping, and elastic deformations. However, these traditional techniques provide limited variability and may not adequately capture the complexity of many datasets.

CNNs applied to these images are prone to overfitting, requiring regularization techniques. For example, Viguera et al. [50] and Busra et al. [51] used methods like Batch Normalization, dropout, and Early Stopping to enhance performance. In recent years, researchers like Wu et al. [52], Sánchez et al. [53], and Fabijańska et al. [1] have advanced semi-supervised learning strategies to extract multi-level features from unlabeled corneal endothelial images, using fine-tuning with minimal labeled data for segmentation.

Moreover, Kucharski and Fabijanska [54] developed a technique using GANs to generate training data for corneal endothelial image segmentation, addressing the scarcity of annotated images. They validated the approach using a UNet model, first trained with labeled images and then with synthetic images from three free web-based databases. The UNet successfully detected mask edges and achieved reasonable generalization. The results, evaluated using accuracy metrics, compared the generated masks with ground truth. The researchers concluded that GAN-based strategies can enhance medical databases with limited annotated images, although GANs are difficult to train, control, and stabilize, requiring extensive adjustments [55].

Similarly, Sierra et al. [46] proposed a segmentation method for specular microscopy images of corneal endothelium affected by Fuchs' dystrophy, framing the task as a regression problem using distance maps rather than traditional pixel-level classification. Their method involves pre-processing to generate distance maps and post-processing to convert them into masks using the watershed technique. Results show faster convergence on clinically relevant parameters, indicating the method's effectiveness with small datasets. However, challenges remain in pre-processing distance maps for morphometric calculations and correcting illumination in the input images.

Fabijańska et al. [4] used a U-NET architecture to segment corneal endothelial images, achieving an AUROC of 0.92 and a DICE coefficient of 0.86, indicating high precision in boundary delineation. Similarly, Nurzynska (2018) applied CNNs for automatic cell segmentation, reaching 93% accuracy compared to manual annotations and a modified Hausdorff distance of 0.14 pixels, demonstrating strong segmentation performance. Hao et al. [56] developed a deep learning system for estimating morphometric parameters and segmenting

corneal endothelial cells from in vivo confocal microscopy images. Tested on 99 subjects, it achieved a Pearson correlation of 0.932 ($p < 0.01$) with Topcon cell density measurements and an AUC of 0.923, surpassing a U-Net model with an AUC of 0.913.

Shilpashree et al. [49] compared average perimeter length (APL) and endothelial cell density (ECD) between Fuchs' endothelial dystrophy (FECD) patients and healthy subjects. Using U-Net and Watershed for segmentation, they found decreased ECD and increased APL with higher guttae in FECD, indicating endothelial deterioration. For healthy subjects, the method achieved an F1 score of 82.27%, an average IoU of 77.27%, a precision of 87.9%, and an ROC AUC of 96.70%, demonstrating high segmentation accuracy.

Hence, there is a compelling need to introduce a data augmentation approach that uses deformations to effectively represent the inherent variability within corneal endothelium images, ultimately enhancing the segmentation task in semi-supervised scenarios.

## Materials and methods

In this section, we present the proposed methodology to improve semi-supervised segmentation. Our approach focuses on the natural variability present in corneal endothelial images through controlled and realistic deformations, thus enhancing the robustness and generalizability of segmentation models. The dataset was accessed and used for research purposes between March 2023 and November 2023.

### Dataset description

A set of 90 in vivo specular microscopy images of endothelial cells in two-channel TIF format, with a resolution of 640x480 pixels, obtained from 66 patients with healthy (42 from the right eye) and dystrophic (48 from the left eye) corneas were used. The data was split by patient, it is important to note that there was no overlap of patient images in the training, testing and validation sets. From these images, 271 patches of size 96x96 were extracted that were labeled by expert personnel, and 1719 patches of the same dimensions without annotations, with balanced distributions for all possible cases. Images of the same patient are not repeated in the training set, nor in the test and validation sets. Images were acquired with a Topcon SP3000P specular microscope equipped with Cell Count software. The study protocol received approval from the ethics committee of the Technological University of Bolívar in Colombia. Due to the retrospective approach of the study, the requirement to obtain informed consent was not applicable. Furthermore, the study was carried out in accordance with the principles established in the Declaration of Helsinki.

### Semi-supervised model

We propose a semi-supervised model consisting of two stages. In the first stage, unsupervised learning is used to train various encoders, including ResNet50, ResNet101, DenseNet121, and ResNet101ViT, within a Siamese network architecture based on the Barlow Twins method [57]. This process learns feature representations from a large set of unlabeled images. In the second stage, the encoder weights obtained from the first stage are frozen, and fine-tuning is performed with supervised learning to segment corneal endothelial images using a small number of labeled images.

As shown in Fig 2, the model processes unlabeled images ($X$) through a Siamese network, where each image undergoes geometric transformations (e.g., rotations, flipping, cropping) to generate new samples ($Y^A$ and $Y^B$). These transformed images are passed through identical encoders to produce feature maps ($Z^A$ and $Z^B$), and consistent patterns are identified using the cross-correlation function. The learned weights are then fixed and applied during fine-tuning.

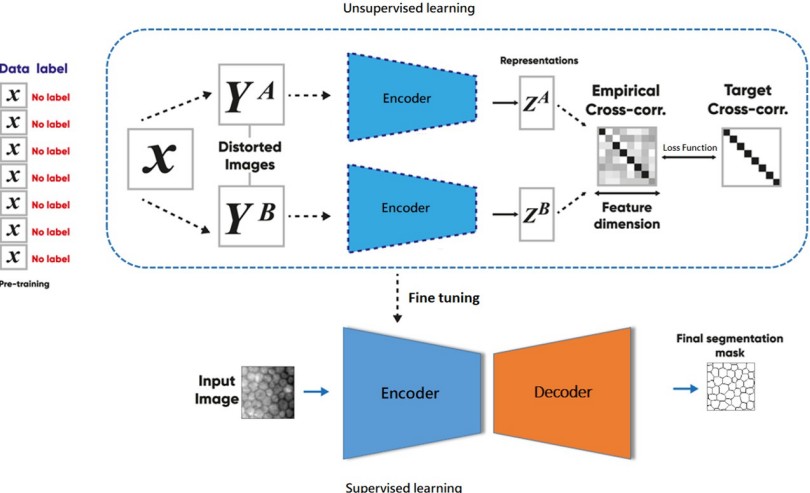

**Fig 2. Block diagram of the proposed semi-supervised model.**

A decoder with attention modules and skip connections is added, and a 1×1 convolution at the output generates the final segmentation mask.

**Architectures.** We use state-of-the-art pretrained backbones—ResNet50 [58], ResNet101 [59], DenseNet121 [60], and ResNet101ViT [61]—within our semi-supervised learning framework. These models, trained on large-scale datasets, have proven effective at learning robust features that enhance performance in our corneal endothelial image segmentation task.

ResNet architectures are used for their deep learning capabilities and residual blocks that prevent performance degradation in deep networks. DenseNet is employed for its dense layer connections, allowing the model to capture detailed and contextual features. Finally, ResNet101ViT, which combines ResNet with Vision Transformer (ViT), captures both local and global features by applying attention mechanisms and processing images in patches [62]. This combination improves feature extraction and segmentation accuracy in medical imaging tasks.

Our approach demonstrates significant improvements in model generalization across different encoders, regardless of the pretrained backbone used.

**Conventional data augmentation.** When there are not enough images to adequately train a model, or the classes have a large imbalance between them, the application of geometric transformations is the simplest way to extend a dataset. However, not all transformations are suitable for this purpose [20]. For this type of data augmentation, this strategy applies a distortion to the input images randomly before the training stage, using a Siamese network using some operations like horizontal flip, vertical flip and rotations.

This strategy can increase the variability of the data set and improve the model's ability to better generalize to new images under different conditions. However, these geometric transformations may not be sufficient to capture the complexity and diversity present in medical data sets.

## Data augmentation via warping

The proposed warping method enhances data augmentation for corneal endothelium images by producing new images based on carefully constructed multistage transformation. The

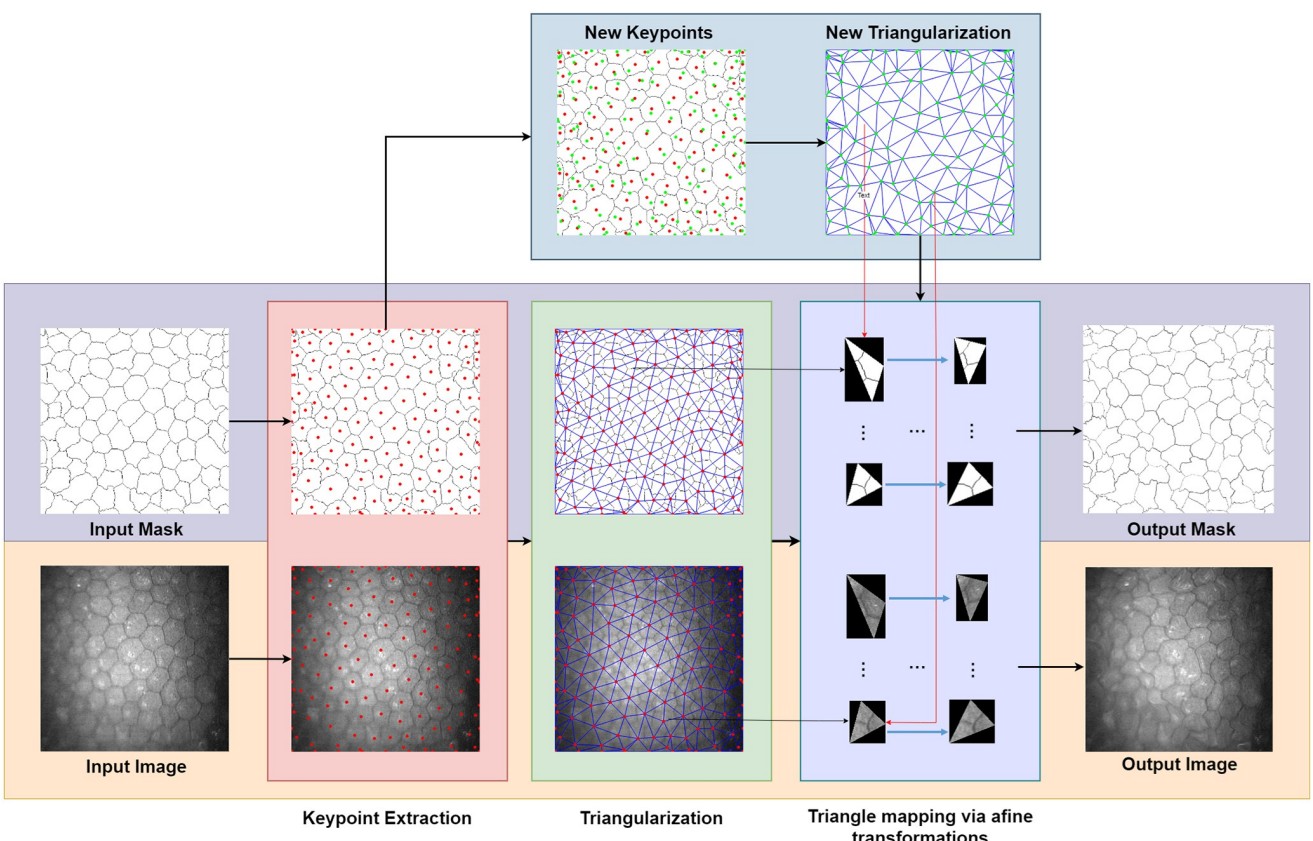

**Fig 3. Proposed data augmentation strategy.**

method takes as input the Image *I* and its segmentation mask *M*. It involves keypoints extraction, Delaunay triangulation, and triangle mapping via affine transformations as outlined in Fig 3.

In Fig 3, you can see that the proposed data augmentation method receives as input images and masks of the corneal endothelium, then these are processed using the warping method, which will be explained later in greater detail.

**Keypoint extraction.** Keypoint extraction is a pivotal step in our warping process, applied to the segmentation mask *M* of the image *I*. This process is designed to capture crucial features from both the internal structure and the edges of the image, ensuring a comprehensive and realistic transformation.

Initially, keypoints are extracted from the mask *M* by identifying the centroids of cells. These points represent significant anatomical features within the corneal endothelium and are essential for guiding the warping transformations. Additionally, to mitigate potential edge artifacts, keypoints are also densely sampled along the image edges. This includes points from critical locations such as the corners, center points, and at intervals along the quarters and eighths of the image dimensions [63].

The final set of keypoints *K* thus encompasses a combination of these two types of extracted points, ensuring both the internal and boundary regions of the image are adequately

represented. The set $K$ is defined as

$$K = \{(x_i, y_i) \mid x_i, y_i \text{ are coordinates of keypoints }\}. \tag{1}$$

This set of keypoints $K$ is instrumental in the subsequent Delaunay triangulation and affine transformations, facilitating the generation of realistic and varied warped images. $x_i$, $y_i$ are the coordinates of the i-th keypoint.

The extraction of closely spaced keypoints near the image edges prevents unrealistic transformations, such as black triangles, as illustrated in Fig 3. This approach ensures deformations are constrained to avoid unrealistic artifacts while still producing augmented training samples.

**Delaunay triangulation.** Delaunay triangulation is used to partition the image into triangular regions, a crucial step for effective warping. This technique is selected for its distinctive properties that are especially beneficial in image processing tasks. Delaunay triangulation maximizes the minimum angle of all triangles, preventing the formation of elongated or skinny triangles, which results in more uniform and stable transformations during the warping process [64]. Moreover, it guarantees a unique set of triangles for a given set of points (assuming no four points are cocircular), providing consistency and predictability essential for repeatable image warping.

Given the set of keypoints $K$, Delaunay triangulation creates a network of triangles, represented mathematically as

$$T = \{\triangle k_a k_b k_c \mid k_a, k_b, k_c \in K, a \neq b \neq c\} \tag{2}$$

where $\triangle k_a k_b k_c$ denotes a triangle formed by keypoints $k_a$, $k_b$, $k_c$. This structured approach ensures that the transformations applied in the warping process are evenly distributed and align with the image's natural geometry.

The choice of Delaunay triangulation is particularly advantageous for image warping. Its ability to adapt to the local geometry of the keypoints provides an optimal tessellation that minimizes potential warping artifacts and distortions. This leads to more natural and visually pleasing results, an essential factor in applications like medical imaging, where maintaining the integrity and accuracy of structural representations is paramount.

**Warping through local affine transformations.** The warping stage transforms the original image $I$ with coordinates $(x, y)$ into the warped image $I_w$ with coordinates $(x', y')$. This transformation is facilitated by local affine transformations applied within triangular regions.

Original keypoints $K$ are modified to create a new set of keypoints $K'$, facilitating the generation of a new triangulation grid for the warped image (Fig 3). Each keypoint with coordinates $(x_i, y_i)$ in $K$ is shifted by random amounts to produce the new keypoints $K'$ given by

$$K' = \{(x_i', y_i') \mid x_i' = x_i + \delta x, \ y_i' = y_i + \delta y\}, \tag{3}$$

where $\delta x$ and $\delta y$ are random shifts. Let $R$ be a random variable following a uniform distribution in $(-1, 1) \in \mathbb{R}$, a random shift, $\delta$, is then computed as follows,

$$\delta = \frac{R \cdot d_{min}}{s}, \tag{4}$$

where $d_{\min}$ is the minimun distance between any two points in $K$ defined as

$$d_{\min} = \min_{a,b \in K, a \neq b} d(a, b), \tag{5}$$

and

$$d(a, b) = \min(|x_a - x_b|, |y_a - y_b|). \tag{6}$$

**Fig 4. Transformation via warping with low ($s = 3$) and high ($s = 2$) deformation levels.**

The parameter $s$ in Eq 4 inversely controls the magnitude of the transformation: a smaller $s$ results in larger shifts and consequently more pronounced warping [65].

After re-triangulation using the modified set $K'$, we obtain a new set of triangles that map the original image to the warped image. The continuous vector-valued warping function $\mathbf{f}$ is defined for each triangle, projecting the pixels from $I$ to $I_w$ given by

$$I_w(x', y') = I(\mathbf{f}^{-1}(x', y')), \quad \forall (x', y') \in \triangle', \tag{7}$$

where $\triangle'$ is the domain of the triangles in $I_w$, and $\mathbf{f}^{-1}(x', y')$ gives the corresponding coordinates in the original image $I$.

This process ensures a bijective and smooth transition from $I$ to $I_w$, resulting in a warped image $I_w$ and warped mask $M_w$ that exhibit realistic variations while maintaining the structural integrity of the original image. The local affine transformations are carefully computed to preserve the topology and avoid artifacts.

Transformations via warping help to enrich the training data set, introducing realistic variations in the images, which allows improving the generalization of CNN networks. Fig 4 shows data augmentation for input images and masks, using the warping method described above.

From the previous Fig 4, we can see that the proposed model can perform deformations of the original image with different levels, this can be achieved by adjusting the parameter $s$ within the algorithm developed. After obtaining the final deformations, it can be seen that the masks do not present fine details on the edges of the cells and guttae, so it is proposed to use a mask refinement stage.

**Mask refinement.**  At this stage, the watershed technique is used in image processing to adjust the cell segmentation masks (Fig 5). This technique is based on the idea of simulating

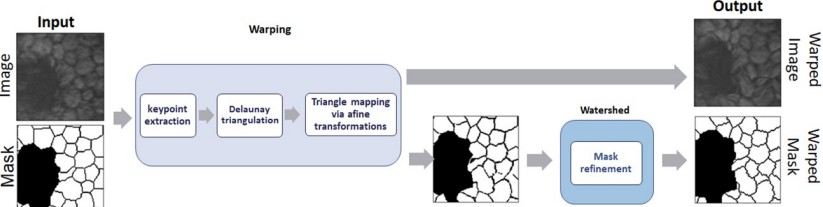

**Fig 5. Block diagram of the proposed strategy.**

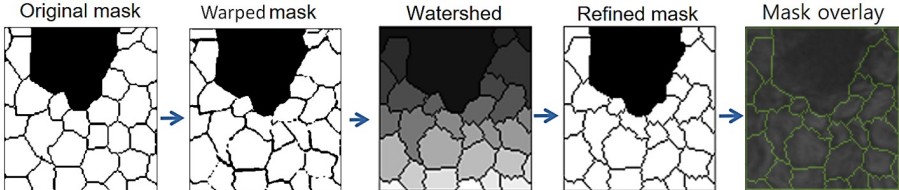

**Fig 6. Edge refinement of a deformed mask of healthy endothelial cells using the watershed technique.**

the behavior of water flowing over a topography, where the points of water accumulation determine the boundaries between cells. By applying a basin to cell images, intensity gradients and regions of overlap are identified, allowing for more precise and detailed segmentation of individual cells. By using the watershed transform, it is possible to adjust the mask edges more effectively, thereby improving the quality of the training data used for AI models, resulting in higher accuracy and performance in cell segmentation [66]. Below is a diagram of the proposed method.

After applying the basin technique to the deformed images, the results obtained can be detailed in Figs 6 and 7.

In the figures above, it can be seen that the basin technique helps to refine the output masks, so that they are as plausible as possible with respect to the original ones. because the masks generated depending on the level of deformation may present incomplete edges, isolated points and very thin or thick cellular regions. Achieving very promising results for medical databases.

## Experiments and results

In this research, a semi-supervised learning model was developed, based on the Barlow Twins approach, with the purpose of improving the performance of CNNs. To train and validate the model, specular microscopy images of the corneal endothelium in vivo with various resolutions were used.

### Experiment configuration

The model was trained in the unsupervised stage with different encoders, using the warmup cosine optimizer with learning rate initialized at $1e^{-3}$ for all experiments, a batch size of 8 and a weight decay $5e^{-4}$. For the Fine Tuning stage to learn the segmentation task, the Adam optimizer and the Binary Cross-Entropy Dice Loss function were used. The feature maps learned in the unsupervised stage were concatenated with a 5-filter decoder [16, 32, 64, 128, 256], which had attention modules, skip connections and residual blocks.

**Fig 7. Edge refinement of a deformed mask of endothelial cells with guttae using the watershed technique.**

**Table 1. Data distribution for training, testing and validation of the semi-supervised model.** (Strain values in parameter s ranged between 1 and 3, with values closer to zero giving more pronounced strains).

| Strategy | Unsupervised Stage (No Labels) | State of Cells in Unlabeled Images | Number of Patients (No Labels) | Supervised Stage (Labels) | State of Cells in Labeled Images | Number of Patients (Labels) |
|---|---|---|---|---|---|---|
| 1 (Baseline) | 1719 | 859 (healthy cells) 860 (cells with dystrophy) | 29 patients (right eye) 29 patients (left eye) | 271 (216 for training, 25 for test, 30 for validation) | Training: 108 (healthy cells) 108 (cells with dystrophy) Test: 12 (healthy cells) 13 (cells with dystrophy) Validation: 15 (healthy cells) 15 (cells with dystrophy) | Training: 4 patients (right eye) 4 patients (left eye) Test: 1 patient (right eye) 1 patient (left eye) Validation: 1 patient (right eye) 1 patient (left eye) |
| 2 | 4220 | 2110 (healthy cells) 2110 (cells with dystrophy) | 29 patients (right eye) 29 patients (left eye) | 271 (216 for training, 25 for test, 30 for validation) | Training: 108 (healthy cells) 108 (cells with dystrophy) Test: 12 (healthy cells) 13 (cells with dystrophy) Validation: 15 (healthy cells) 15 (cells with dystrophy) | Training: 4 patients (right eye) 4 patients (left eye) Test: 1 patient (right eye) 1 patient (left eye) Validation: 1 patient (right eye) 1 patient (left eye) |
| 3 | 4220 | 2110 (healthy cells) 2110 (cells with dystrophy) | 29 patients (right eye) 29 patients (left eye) | 652 (597 for training, 25 for test, 30 for validation) | Training: 298 (healthy cells) 299 (cells with dystrophy) Test: 12 (healthy cells) 13 (cells with dystrophy) Validation: 15 (healthy cells) 15 (cells with dystrophy) | Training: 10 patients (right eye) 10 patients (left eye) Test: 1 patient (right eye) 1 patient (left eye) Validation: 1 patient (right eye) 1 patient (left eye) |

In order to demonstrate the applicability of the proposed data augmentation technique, the semi-supervised model was trained in three scenarios. For the first scenario called Strategy 1 (Baseline), the corneal endothelium dataset with conventional augmentation was used in all training stages of the model (horizontal flip, vertical flip, and rotations), employing 1719 patches without labels for unsupervised training and 271 patches with labels (216 for training, 25 for test and 30 for validation) to learn the segmentation task, performing fine tuning under supervised learning. For the second scenario called Strategy 2, it was increased from 1719 to 4220 unlabeled patches, using the proposed data augmentation technique, for the unsupervised stage. For the last scenario called Strategy 3, the number of images of Strategy 2 was used in the unsupervised stage and for the supervised stage it was increased from 271 to 652 labeled patches, 597 for training, 25 for test and 30 for validation, using the proposed data augmentation. The Table 1 shows the distribution of the data according to the strategy used. And Table 2 shows the quantitative results taking into account several evaluation metrics for the segmentation task.

**Table 2. Quantitative comparative analysis of the semi-supervised model for different data augmentation scenarios using the evaluation metrics coefficient dice (DC), accuracy (Acc), Area Under the Receiver Operating Characteristic Curve (AUROC), and mean intersection-over-union (mIoU).**

| Model | Strategy 1 | | | | Strategy 2 | | | | Strategy 3 | | | |
|---|---|---|---|---|---|---|---|---|---|---|---|---|
| | Auroc | Acc | Dc | mIoU | Auroc | Acc | Dc | mIoU | Auroc | Acc | Dc | mIoU |
| **BT-UNet** | 0.70 | 0.69 | 0.78 | 0.60 | 0.81 | 0.81 | 0.84 | 0.66 | **0.88** | **0.87** | **0.85** | **0.72** |
| **BT-ResNet50** | 0.78 | 0.78 | 0.82 | 0.65 | 0.86 | 0.84 | 0.85 | 0.67 | **0.90** | **0.91** | **0.91** | **0.83** |
| **BT-ResNet101** | 0.63 | 0.63 | 0.83 | 0.66 | 0.83 | 0.84 | 0.86 | 0.65 | **0.87** | **0.89** | **0.91** | **0.85** |
| **BT-ResNet101ViT** | 0.78 | 0.76 | 0.82 | 0.67 | 0.83 | 0.83 | 0.84 | 0.68 | **0.85** | **0.84** | **0.89** | **0.76** |
| **BT-DenseNet121** | 0.81 | 0.81 | 0.86 | 0.67 | 0.88 | 0.88 | 0.87 | 0.69 | **0.92** | **0.91** | **0.94** | **0.86** |

The sample distribution in Table 1 was designed with careful consideration of the data and study objectives. More samples were allocated to the validation set than the test set to fine-tune hyperparameters during training. This approach improves performance on unseen data and helps prevent overfitting. We also recognize the importance of the test set for obtaining an unbiased estimate of model performance.

During data augmentation, deformations were applied by adjusting the parameter "s" to values greater than zero. Plausible distortions were selected, while higher distortions rendered images unusable, resulting in non-uniform augmentation. This increased the unsupervised training data from 1719 to 4220 (2.76-fold) and the supervised data from 216 to 597 (2.45-fold). Greater augmentation was applied in the unsupervised stage (Barlow twins) to capture feature diversity, while less augmentation was used in the supervised stage to avoid overfitting and ensure generalization.

Below in Fig 8 shows the quantitative results taking into account some evaluation metrics for the segmentation task.

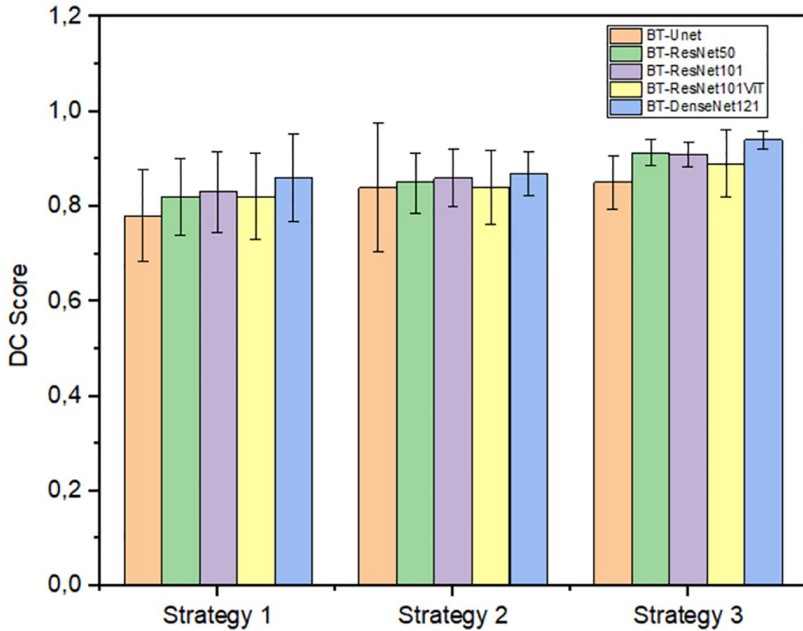

**Fig 8. Quantitative results of the Dice Coefficient metric (DC) using strategy 1 (Baseline), 2 and 3 with the proposed data augmentation method.** The acronym BT stands for Barlow Twins.

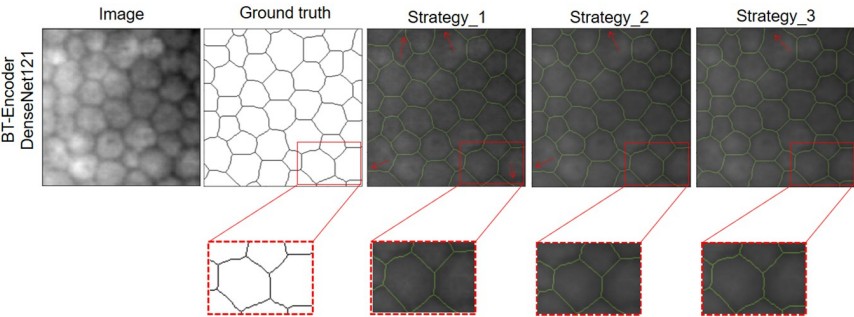

**Fig 9. Segmentation results for an image with healthy cells using the semi-supervised model and the proposed data augmentation method.**

Fig 8 shows the Dice Coefficient (DC) metric calculated for different semi-supervised models. It is evident that the proposed strategy improves the performance of the models significantly compared to the baseline (strategy 1). Where improvements can be seen by implementing data augmentation only in the unsupervised learning stage and more promising results if it is also applied in the fine tuning of the network. Below, in Table 2, the quantitative results of the architectures used can be observed.

From the Table 2 and Fig 8, it can be observed that the best performance is obtained for the case of Strategy 3, which allows us to affirm that the proposed data augmentation process allows the semi-supervised model to generalize better. This is due to the fact that the generated images represent natural and realistic deformations, which help the diversity of the data.

Our semi-supervised model, utilizing the proposed data augmentation method, demonstrates superior performance in segmenting corneal endothelium images, especially in the presence of guttae. The DenseNet121 encoder-based model achieved an AUROC of 0.92, a precision of 0.91, a DICE coefficient of 0.94, and a mean IoU of 0.86.

As shown in Table 1, our model was evaluated using cellular microscopy images of the corneal endothelium, including both healthy cells and those affected by Fuchs dystrophy. This comprehensive evaluation addresses a significant limitation of current strategies, which predominantly focus on healthy cells. The challenge of accurately segmenting cells with guttas in modern specular microscopy systems, where automatic segmentation often fails, was effectively tackled by our approach.

For comparison, Fabijańska et al. [4] reported an AUROC of 0.92 and a DICE coefficient of 0.86 using a U-NET-based network for segmenting healthy corneal endothelial images. Nurzynska (2018) achieved 93% accuracy in overlapping automatic delineation with manual annotations and a modified Hausdorff distance of 0.14 pixels. Hao et al. [56] developed a deep learning system for segmenting corneal endothelial cells, achieving an AUC of 0.923 and a Pearson correlation coefficient of 0.932 for estimated morphometric parameters. Shilpashree et al. [49] reported an F1 Score of 82.27%, a mean IoU of 77.27%, an accuracy of 87.9%, and an AUROC of 96.70% for healthy subjects using a combination of U-Net and Watershed.

Our method surpasses these benchmarks, particularly in handling the variability and challenges posed by diseased cells, demonstrating the efficacy of incorporating data augmentation via warping transforms to enhance semi-supervised segmentation tasks.

In Fig 9, it can be seen that the model receives as input an image with healthy endothelial cells, with the objective of evaluating the prediction of the output mask using a semi-supervised model with DenseNet121 encoder trained with the strategy 1, 2 and 3. Where it can be

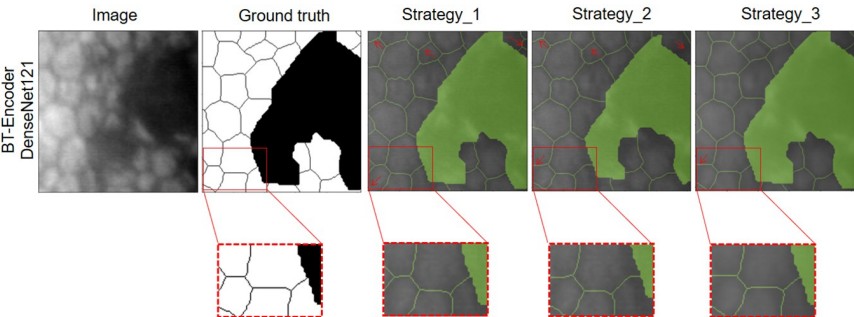

**Fig 10. Results of the segmentation of an endothelial image with the presence of guttae using the semi-supervised model and the proposed data augmentation method.**

seen that strategy 3 presented the best performance having the ground truth as a reference. In Fig 10, the model receives an image with diseased endothelial cells with some lighting challenges, where strategy 3 still has the best precision compared to the other strategies. All the previous results show that the proposed data augmentation strategy helps to improve the generalization of CNN networks.

## Conclusions

Deep learning models, such as CNNs, often face difficulties in terms of generalization and prevention of overfitting when the labeled data set is limited. In this study, we have presented an innovative data augmentation strategy that is based on the warping and watershed method, with the purpose of improving the performance of these algorithms in the task of semantic segmentation of specular microscopy images of the corneal endothelium. The results obtained are highly promising compared to traditional data augmentation techniques, despite the presence of variations in scale, lighting, shadows, brightness, lack of sharpness and other factors inherent in this type of images. These conclusions are supported by implementing three data augmentation experiments using a semi-supervised learning approach, where the proposed method proved to be the most effective. To verify the effectiveness of the predictions, the accuracy, mIoU and Dice coefficient metrics were used. Finally, it can be stated that this technique will be of great help for medical databases that have the limitation of few labeled images and present deformation within their ROIs.

## Acknowledgments

S. Sanchez thanks Minciencias and Sistema General de Regalías (Programa de Becas de Excelencia) for a PhD scholarship.

## Author Contributions

**Conceptualization:** Sergio Sanchez, Noelia Vallez, Andres G. Marrugo.

**Data curation:** Andres G. Marrugo.

**Funding acquisition:** Andres G. Marrugo.

**Investigation:** Sergio Sanchez, Noelia Vallez.

**Methodology:** Sergio Sanchez, Gloria Bueno, Andres G. Marrugo.

**Project administration:** Andres G. Marrugo.

**Resources:** Andres G. Marrugo.

**Software:** Sergio Sanchez, Noelia Vallez.

**Supervision:** Gloria Bueno, Andres G. Marrugo.

**Visualization:** Sergio Sanchez, Noelia Vallez.

**Writing – original draft:** Sergio Sanchez, Noelia Vallez, Andres G. Marrugo.

**Writing – review & editing:** Sergio Sanchez, Noelia Vallez, Gloria Bueno, Andres G. Marrugo.

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
