## [Editor Report · Decision Letter 0]

7 May 2024

PONE-D-24-00975Data augmentation via warping transforms for modeling natural variability in the corneal endothelium enhances semi-supervised segmentationPLOS ONE

Dear Dr. Marrugo,

Thank you for submitting your manuscript to PLOS ONE. After careful consideration, we feel that it has merit but does not fully meet PLOS ONE’s publication criteria as it currently stands. Therefore, we invite you to submit a revised version of the manuscript that addresses the points raised during the review process.

We look forward to receiving your revised manuscript.

Kind regards,

Akram Belghith

Academic Editor

PLOS ONE

Journal Requirements:

 [This work has been partly funded by Ministerio de Ciencia, Tecnología e Innovación, Colombia, Project 124489786239 (Contract 763-2021).].  

[This work has been partly funded by Ministerio de Ciencia, Tecnolog´ıa e Innovaci´on,

Colombia, Project 124489786239 (Contract 763-2021). S. Sanchez thanks Minciencias

and Sistema General de Regal´ıas (Programa de Becas de Excelencia) for a PhD

scholarship]

 [This work has been partly funded by Ministerio de Ciencia, Tecnología e Innovación, Colombia, Project 124489786239 (Contract 763-2021).]

6. This study explores advancements in the segmentation of corneal endothelium images through a novel data augmentation approach. The authors propose a semi-supervised learning framework that integrates unsupervised data with a smaller subset of labeled data to improve segmentation accuracy. Utilizing a combination of models including DenseNet121, and a unique warping transformation technique, the paper claims to outperform standard augmentation methods. While the findings are significant as they offer the potential for more detailed segmentation, the paper requires major revisions in data interpretation and evaluation of results.

Additional Editor Comments 

Related work:

1.     Since, the article aims to enhance the segmentation algorithm by improving data augmentation, it is recommended that the authors provide this with numerical results from previous related works for clearer benchmarking in this section (AUC, ACC and Dice)

Dataset description and Table 1:

2.     The authors should specify the number of unique patients and eyes in each dataset (training, testing, and validation). Additionally, details on the distribution of healthy and pathological corneas in these sets would be valuable.

3.     Authors should clarify whether data was split by image, eye, or patient level and confirm whether there is any overlap of images from the same patient across the training, testing, and validation datasets.

4.    Table 1: The allocation of data into 216 training, 25 testing, and 30 validation samples deviates from conventional norms, particularly with the validation set being larger than the testing set. Could the authors provide a rationale for this distribution?

5.     Table 1: The article describes using data augmentation to increase the sample size. Typically, augmentation is applied uniformly across all images, and they are either replaced or combined with the original images, resulting in the augmented dataset being an integer multiple of the original dataset. However, Table 1 shows that the train data increased from 1719 to 4220 (a 2.76 times increase) and from 216 to 597 (a 2.45 times increase) for the unsupervised and supervised sections, respectively. This is not consistent with an integer multiple increase. The authors need to clarify whether the augmentation was applied selectively to parts of the data multiple times or if another method was used.

Semi Supervised Model and Architecture:

6.     The authors should provide clearer details about the unsupervised model within the semi-supervised framework shown in Figure 2. Specifically, clarify whether Models ResNet50, ResNet101, DenseNet121, and ResNet101ViT are used as encoders in the unsupervised section, and confirm if DenseNet121 is the only encoder used in the supervised section, or if the same models are employed across both sections. Please revise this section (and Figure 2 if necessary) for greater clarity.

Data Augmentation section:

7.     In Figure 3, during the keypoints extraction phase, the method extracts points at equal distances along all borders of the image (including all four sides and each square). However, these points do not appear to be centered on any cell. As a result, some cells, especially on the edges, have two very closely spaced centers identified, one of which may be incorrect. This redundancy could significantly impact the performance of the Delaunay triangulation. Could the authors please clarify this issue?

8.   In The section "Warping through Local Affine Transformations" discusses the parameter   and its influence on image deformation, yet  is absent from the formulas provided. Additionally, it is unclear which numerical values of  correspond to the data in Table 1 (are these values from high- or low-deformation categories used in the training dataset?). The authors need to specify the numerical values of  used in Table 1 and Figure 4 to help readers understand this parameter's impact and efficacy.

Results and conclusions:

9.     The names of the models prefixed with "BT-" in Table 2 and Figure 8 are not defined in the text. Please clarify what "BT-" stands for.

10.  The authors should include the AUROC as a metric in table 2, alongside ACC. AUROC is more informative for segmenting corneal endothelium images, where class imbalances are common (most pixels being non-border, class 0) and can provide a more comprehensive evaluation of the model's performance across various thresholds.

11.  The authors should compare and discuss their results with those of other methods that have shown superior performance in this domain. For example, the study linked here https://pubmed.ncbi.nlm.nih.gov/29680687/ achieved an AUROC of 0.92 using conventional augmentation methods.

12.  The authors should add the results from common augmentation methods to Table 2 for comparison with the proposed method. This will help determine how much of the improvement is due to the unique and effective augmentation approach introduced in the article.

---

## [Author Response · Author response to Decision Letter 0]

3 Jul 2024

PLOS ONE Manuscript: PONE-D-24-00975

Data augmentation via warping transforms for modeling natural variability in the corneal endothelium enhances semi-supervised segmentation

Dear Dr. Belghith,

We are submitting our response letter to the reviewer comments and the revised manuscript. Overall, we appreciate the constructive review, which has led to an improved paper. We want to make the peer review publicly available. Also, we have made the data and the analysis code available on a public repository. The conventions we have used in this letter are: 

Black text in italics are editor or reviewer comments

Blue text are our responses to those comments

Red text are the new or modified text in the manuscript. When necessary we have included a figure from the paper here or a screenshot with equations.

Regards,

Andrés Marrugo, Ph.D.

Professor of Mechatronics Engineering

Universidad Tecnológica de Bolívar

Cartagena, Colombia

---

Journal Requirements:

1. When submitting your revision, we need you to address these additional requirements. Please ensure that your manuscript meets PLOS ONE's style requirements, including those for file naming. The PLOS ONE style templates can be found at https://journals.plos.org/plosone/s/file?id=wjVg/PLOSOne_formatting_sample_main_body.pdf and https://journals.plos.org/plosone/s/file?id=ba62/PLOSOne_formatting_sample_title_authors_affiliations.pdf. 

RE: The manuscript has been adjusted to the style requirements of PLOS ONE.

RE: The code resulting from the research is available in the following Github repository and the link has been added to the Data availability statement (line 425) at the end of the manuscript: https://github.com/checho9214/Data-augmentation-via-warping---semi-supervised-segmentation

3. Thank you for stating the following financial disclosure: [This work has been partly funded by Ministerio de Ciencia, Tecnología e Innovación, Colombia, Project 124489786239 (Contract 763-2021).]. Please state what role the funders took in the study. If the funders had no role, please state: ""The funders had no role in study design, data collection and analysis, decision to publish, or preparation of the manuscript.""

If this statement is not correct you must amend it as needed. Please include this amended Role of Funder statement in your cover letter; we will change the online submission form on your behalf.

RE: Thank you for the recomendation. We have updated our funding statement on the cover letter and we would like to update our funding statement as follows: 

This work has been partly funded by Ministerio de Ciencia, Tecnología e Innovación, Colombia, Project 124489786239 (Contract 763-2021). S. Sanchez thanks Minciencias and Sistema General de Regalías (Programa de Becas de Excelencia) for a PhD scholarship. The funders had no role in study design, data collection and analysis, decision to publish, or preparation of the manuscript.

4. Thank you for stating the following in the Acknowledgments Section of your manuscript: [This work has been partly funded by Ministerio de Ciencia, Tecnologıa e Innovacion, Colombia, Project 124489786239 (Contract 763-2021). S. Sanchez thanks Minciencias and Sistema General de Regal´ıas (Programa de Becas de Excelencia) for a PhD scholarship] We note that you have provided funding information that is not currently declared in your Funding Statement. However, funding information should not appear in the Acknowledgments section or other areas of your manuscript. We will only publish funding information present in the Funding Statement section of the online submission form. Please remove any funding-related text from the manuscript and let us know how you would like to update your Funding Statement. Currently, your Funding Statement reads as follows: [This work has been partly funded by Ministerio de Ciencia, Tecnología e Innovación, Colombia, Project 124489786239 (Contract 763-2021).]

RE: We have modified the Acknowledgments section accordingly.

RE: Thanks for your recomendation. We have moved the ethics statement to the methods section in line 213.

6. This study explores advancements in the segmentation of corneal endothelium images through a novel data augmentation approach. The authors propose a semi-supervised learning framework that integrates unsupervised data with a smaller subset of labeled data to improve segmentation accuracy. Utilizing a combination of models including DenseNet121, and a unique warping transformation technique, the paper claims to outperform standard augmentation methods. While the findings are significant as they offer the potential for more detailed segmentation, the paper requires major revisions in data interpretation and evaluation of results.

RE: Thank you for your recommendation. We have added more details on related work (page 5) for a more nuanced data interpretation and evaluation of results, and have added further discussion and interpretation in pages 12 and 13 with an additional performance metric (AUROC).

Additional Editor Comments 

Related work:

Since, the article aims to enhance the segmentation algorithm by improving data augmentation, it is recommended that the authors provide this with numerical results from previous related works for clearer benchmarking in this section (AUC, ACC and Dice)

RE: Thank you for this suggestion. We have added other performance metrics and have also reported these metrics as a benchmark for our proposed method. Nevertheless, it is important to note that most work in the literature reports performance based on healthy cells. We added this analysis and discussion in page 13. We finish this discussion with the following:

Our method surpasses these benchmarks, particularly in handling the variability and challenges posed by diseased cells, demonstrating the efficacy of incorporating data augmentation via warping transforms to enhance semi-supervised segmentation tasks.

Dataset description and Table 1:

The authors should specify the number of unique patients and eyes in each dataset (training, testing, and validation). Additionally, details on the distribution of healthy and pathological corneas in these sets would be valuable.

RE: Thank you for your recommendation. We have revised the data description to the following in page 6:

A set of 90 in vivo specular microscopy images of endothelial cells in two-channel TIF format, with a resolution of 640x480 pixels, obtained from 66 patients with healthy (42 from the right eye) and dystrophic (48 from the left eye) corneas were used. The data was split by patient, it is important to note that there was no overlap of patient images in the training, testing and validation sets. From these images, 271 patches of size 96x96 were extracted that were labeled by expert personnel, and 1719 patches of the same dimensions without annotations, with balanced distributions for all possible cases. Images of the same patient are not repeated in the training set, nor in the test and validation sets. Images were acquired with a Topcon SP3000P specular microscope equipped with Cell Count software. The study protocol received approval from the ethics committee of the Technological University of Bolívar in Colombia. Due to the retrospective approach of the study, the requirement to obtain informed consent was not applicable. Furthermore, the study was carried out in accordance with the principles established in the Declaration of Helsinki.

Authors should clarify whether data was split by image, eye, or patient level and confirm whether there is any overlap of images from the same patient across the training, testing, and validation datasets.

RE: Thank you for your recommendation. The data were divided by patients, a set of 90 in vivo specular microscopy images of endothelial cells in two-channel TIF format, with a resolution of 640x480 pixels, obtained from 66 patients with healthy corneas (42 from the right eye) and dystrophic corneas (48 of the left eye). In the case of the images of diseased cells with Fuchs' dystrophy, low, medium and high levels of gutta were observed, distributed from the center of the cornea to the periphery. Importantly, there was no overlap of patient images in the training, testing, and validation sets.

Table 1: The allocation of data into 216 training, 25 testing, and 30 validation samples deviates from conventional norms, particularly with the validation set being larger than the testing set. Could the authors provide a rationale for this distribution?

RE: Thank you for your recommendation. The distribution of samples in Table 1 reflects careful consideration of our data's nature and our study's objectives. We assigned more samples to the validation set than the testing set to fine-tune model hyperparameters during training. This choice helps improve model performance on unseen data and prevents overfitting to the training set. We acknowledge the regulations and the importance of testing data to obtain an unbiased estimate and verify the model's performance in practice. The difference of 5 images between the validation and testing stages is not significant, but validation is crucial for fine-tuning the model and avoiding overfitting.

Table 1: The article describes using data augmentation to increase the sample size. Typically, augmentation is applied uniformly across all images, and they are either replaced or combined with the original images, resulting in the augmented dataset being an integer multiple of the original dataset. However, Table 1 shows that the train data increased from 1719 to 4220 (a 2.76 times increase) and from 216 to 597 (a 2.45 times increase) for the unsupervised and supervised sections, respectively. This is not consistent with an integer multiple increase. The authors need to clarify whether the augmentation was applied selectively to parts of the data multiple times or if another method was used.

RE: Thank you for your recommendation. During data augmentation, we generated different deformations for each image by adjusting the parameter "s" to values greater than zero. Distortions larger than zero were more plausible and could be selected, while higher distortions often rendered images implausible and unusable. Therefore, the augmentation process was not uniform. This resulted in the training data increasing from 1719 to 4220 (a 2.76-fold increase) for the unsupervised section and from 216 to 597 (a 2.45-fold increase) for the supervised section.

We applied data augmentation in different proportions to the unsupervised and supervised stages. The unsupervised stage (Barlow twins) required greater variability to capture the diversity of features, hence the higher augmentation. In the supervised stage, where segmentation is performed, a less pronounced augmentation was applied to avoid overfitting and ensure proper generalization to unseen data.

Semi Supervised Model and Architecture:

The authors should provide clearer details about the unsupervised model within the semi-supervised framework shown in Figure 2. Specifically, clarify whether Models ResNet50, ResNet101, DenseNet121, and ResNet101ViT are used as encoders in the unsupervised section, and confirm if DenseNet121 is the only encoder used in the supervised section, or if the same models are employed across both sections. Please revise this section (and Figure 2 if necessary) for greater clarity.

RE: In response to this critique, we propose a weakly supervised model implemented in two stages. In the first stage, unsupervised learning uses different backbones (ResNet50, ResNet101, DenseNet121, and ResNet101ViT) as encoders pre-trained with a Siamese network based on the Barlow Twins method. This stage aims to learn feature representations without data annotations. In the second stage, these pre-trained weights are frozen, and fine-tuning is performed with supervised learning to learn the segmentation task using a few images of the corneal endothelium. Below is the block diagram of the proposed architecture.

Figure 2: Block diagram of the proposed semi-supervised model

Data Augmentation section:

In Figure 3, during the keypoints extraction phase, the method extracts points at equal distances along all borders of the image (including all four sides and each square). However, these points do not appear to be centered on any cell. As a result, some cells, especially on the edges, have two very closely spaced centers identified, one of which may be incorrect. This redundancy could significantly impact the performance of the Delaunay triangulation. Could the authors please clarify this issue?

RE: Thank you for your recommendation. The extraction of closely spaced keypoints near the image edges prevents unrealistic transformations, such as black triangles. While this approach limits the maximum deformation for the image, it still allows for the generation of new training images. This method ensures deformations are constrained to avoid unrealistic transformations, yet sufficient to produce augmented training samples. We added the following comment in page 9:

The extraction of closely spaced keypoints near the image edges prevents unrealistic transformations, such as black triangles, as illustrated in fig. 3. This approach ensures deformations are constrained to avoid unrealistic artifacts while still producing augmented training samples. 

In The section "Warping through Local Affine Transformations" discusses the parameter and its influence on image deformation, yet is absent from the formulas provided. Additionally, it is unclear which numerical values of correspond to the data in Table 1 (are these values from high- or low-deformation categories used in the training dataset?). The authors need to specify the numerical values of used in Table 1 and Figure 4 to help readers understand this parameter's impact and efficacy.

RE: Thank you for pointing this out. We have further explained how the warping parameter s in the manuscript and have updated the table and figure to clarify the values used and the impact in the final transformation as follows:

Results and conclusions:

The names of the models prefixed with "BT-" in Table 2 and Figure 8 are not defined in the text. Please clarify what "BT-" stands for.

RE: Thank you for your recommendation. BT is the contraction of the word Barlow Twins. The legend of this meaning was added in figure 8.

The authors should include the AUROC as a metric in table 2, alongside ACC. AUROC is more informative for segmenting corneal endothelium images, where class imbalances are common (most pixels being non-border, class 0) and can provide a more comprehensive evaluation of the model's performance across various thresholds.

RE: Thank you for this suggestion. We have included the AUROC metric and all models were trained again to compute the performance metrics obtaining satisfactory results compared to the state of the art. As can be seen in the following table. This has been described further in page 13.

From the above table, we can see that the data augmentation strategy with warping method and semi-supervised model is an excellent option when little labeled medical data is available. This approach shows good performance for both imaging healthy cells and those with Fuchs dystrophy. Strategy 3 presented the best performan

---

## [Decision Letter · Decision Letter 1]

12 Sep 2024

PONE-D-24-00975R1Data augmentation via warping transforms for modeling natural variability in the corneal endothelium enhances semi-supervised segmentationPLOS ONE

Dear Dr. Marrugo,

Thank you for submitting your manuscript to PLOS ONE. After careful consideration, we feel that it has merit but does not fully meet PLOS ONE’s publication criteria as it currently stands. Therefore, we invite you to submit a revised version of the manuscript that addresses the points raised during the review process.

We look forward to receiving your revised manuscript.

Kind regards,

Akram Belghith

Academic Editor

PLOS ONE

**Journal Requirements:**

Reviewers' comments:

Reviewer's Responses to Questions

**Comments to the Author**

1. If the authors have adequately addressed your comments raised in a previous round of review and you feel that this manuscript is now acceptable for publication, you may indicate that here to bypass the “Comments to the Author” section, enter your conflict of interest statement in the “Confidential to Editor” section, and submit your "Accept" recommendation.

Reviewer #1: (No Response)

2. Is the manuscript technically sound, and do the data support the conclusions?

Reviewer #1: (No Response)

3. Has the statistical analysis been performed appropriately and rigorously? 

Reviewer #1: (No Response)

4. Have the authors made all data underlying the findings in their manuscript fully available?

Reviewer #1: (No Response)

5. Is the manuscript presented in an intelligible fashion and written in standard English?

Reviewer #1: (No Response)

6. Review Comments to the Author

**Reviewer #1: **Comment 1: The authors are encouraged to succinctly summarize the newly added related work, focusing particularly on the essential details relevant to the study, as cited in References 57-59.

Comment 2: Please ensure that all abbreviations are clearly defined at their first occurrence within the manuscript. For example, "The UW approach" has not been defined, which I presume might refer to 'U-Net and Watershed.' This abbreviation should be explicitly clarified to maintain clarity for all readers.

Comment 3: While the added "Dataset Description" section is informative, Table 1 currently lacks sufficient details regarding the distribution of data across the train, validation, and test sets. I recommend enhancing Table 1 with additional columns to clarify the following aspects:

Patient Distribution: Out of the 66 unique patients in this study, please specify how many patients belong to the healthy category and how many to the dystrophic category.

Dataset Breakdown: For each category (healthy and dystrophic), provide details on the number of unique patients included in the train, validation, and test datasets (e.g., the training set includes 23 unique patients with 17 being healthy and 6 dystrophic).

Specular Microscopy Images: Apply the same detailed breakdown to the 90 in vivo specular microscopy images regarding their distribution across the train, validation, and test datasets.

Additionally, please revise the newly added paragraph to reflect these details for enhanced clarity and completeness.

Comment 4: Please revise and incorporate the main points from your following responses in "Author's Response To Reviewer Comments" file (answers for different comments) into the discussion section of your manuscript:

4.1:

"The distribution of samples in Table 1 reflects careful consideration of our data's nature and our study's objectives. We assigned more samples to the validation set than the testing set to fine-tune model hyperparameters during training. This choice helps improve model performance on unseen data and prevents overfitting to the training set. We acknowledge the regulations and the importance of testing data to obtain an unbiased estimate and verify the model's performance in practice."

4.2:

"During data augmentation, we generated different deformations for each image by adjusting the parameter "s" to values greater than zero. Distortions larger than zero were more plausible and could be selected, while higher distortions often rendered images implausible and unusable. Therefore, the augmentation process was not uniform. This resulted in the training data increasing from 1719 to 4220 (a 2.76-fold increase) for the unsupervised section and from 216 to 597 (a 2.45-fold increase) for the supervised section.

We applied data augmentation in different proportions to the unsupervised and supervised stages. The unsupervised stage (Barlow twins) required greater variability to capture the diversity of features, hence the higher augmentation. In the supervised stage, where segmentation is performed, a less pronounced augmentation was applied to avoid overfitting and ensure proper generalization to unseen data."

Comment 5: The authors' response regarding the "Semi-Supervised Model and Architecture" is notably concise and effectively clarifies the methods used in their research, despite being significantly shorter than the corresponding section in the manuscript. I recommend revising the "Architectures" section to mirror this clarity and conciseness, which will enhance the overall readability and precision of the manuscript.

7. PLOS authors have the option to publish the peer review history of their article (what does this mean?). If published, this will include your full peer review and any attached files.

Reviewer #1: **Yes: **Jalil Jalili

---

## [Author Response · Author response to Decision Letter 1]

22 Sep 2024

Reviewer #1 

Comment 1: The authors are encouraged to succinctly summarize the newly added related work, focusing particularly on the essential details relevant to the study, as cited in References 57-59.

RE: We appreciate your suggestion. We have summarized these paragraphs further.

Fabijańska et al. \\cite{PMID:29680687} used a U-NET architecture to segment corneal endothelial images, achieving an AUROC of 0.92 and a DICE coefficient of 0.86, indicating high precision in boundary delineation. Similarly, Nurzynska (2018) applied CNNs for automatic cell segmentation, reaching 93\\% accuracy compared to manual annotations and a modified Hausdorff distance of 0.14 pixels, demonstrating strong segmentation performance. Hao et al. \\cite{QU2022142} developed a deep learning system for estimating morphometric parameters and segmenting corneal endothelial cells from in vivo confocal microscopy images. Tested on 99 subjects, it achieved a Pearson correlation of 0.932 ($p<0.01$) with Topcon cell density measurements and an AUC of 0.923, surpassing a U-Net model with an AUC of 0.913.

Shilpashree et al. \\cite{shilpashree2021automated} compared average perimeter length (APL) and endothelial cell density (ECD) between Fuchs' endothelial dystrophy (FECD) patients and healthy subjects. Using U-Net and Watershed for segmentation, they found decreased ECD and increased APL with higher guttae in FECD, indicating endothelial deterioration. For healthy subjects, the method achieved an F1 score of 82.27\\%, an average IoU of 77.27\\%, a precision of 87.9\\%, and an ROC AUC of 96.70\\%, demonstrating high segmentation accuracy.

Comment 2: Please ensure that all abbreviations are clearly defined at their first occurrence within the manuscript. For example, "The UW approach" has not been defined, which I presume might refer to 'U-Net and Watershed.' This abbreviation should be explicitly clarified to maintain clarity for all readers.

RE: Thank you for the recommendation. We have verified that all abbreviations are defined. The "UW approach" has been rewritten to avoid the unnecessary acronym.

Comment 3: While the added "Dataset Description" section is informative, Table 1 currently lacks sufficient details regarding the distribution of data across the train, validation, and test sets. I recommend enhancing Table 1 with additional columns to clarify the following aspects:

Patient Distribution: Out of the 66 unique patients in this study, please specify how many patients belong to the healthy category and how many to the dystrophic category.

Dataset Breakdown: For each category (healthy and dystrophic), provide details on the number of unique patients included in the train, validation, and test datasets (e.g., the training set includes 23 unique patients with 17 being healthy and 6 dystrophic).

Specular Microscopy Images: Apply the same detailed breakdown to the 90 in vivo specular microscopy images regarding their distribution across the train, validation, and test datasets.

Additionally, please revise the newly added paragraph to reflect these details for enhanced clarity and completeness.

RE: Thank you for your valuable suggestion. In response, we have enhanced Table 1 to include detailed information regarding the distribution of unique patients (healthy and dystrophic) and the breakdown of the images across the training, validation, and test datasets. 

Comment 4: Please revise and incorporate the main points from your following responses in "Author's Response To Reviewer Comments" file (answers for different comments) into the discussion section of your manuscript:

4.1:

 "The distribution of samples in Table 1 reflects careful consideration of our data's nature and our study's objectives. We assigned more samples to the validation set than the testing set to fine-tune model hyperparameters during training. This choice helps improve model performance on unseen data and prevents overfitting to the training set. We acknowledge the regulations and the importance of testing data to obtain an unbiased estimate and verify the model's performance in practice."

4.2:

"During data augmentation, we generated different deformations for each image by adjusting the parameter "s" to values greater than zero. Distortions larger than zero were more plausible and could be selected, while higher distortions often rendered images implausible and unusable. Therefore, the augmentation process was not uniform. This resulted in the training data increasing from 1719 to 4220 (a 2.76-fold increase) for the unsupervised section and from 216 to 597 (a 2.45-fold increase) for the supervised section.

We applied data augmentation in different proportions to the unsupervised and supervised stages. The unsupervised stage (Barlow twins) required greater variability to capture the diversity of features, hence the higher augmentation. In the supervised stage, where segmentation is performed, a less pronounced augmentation was applied to avoid overfitting and ensure proper generalization to unseen data."

RE: Thank you for your valuable suggestions. In response, the key points from our previous responses to comments 4.1 and 4.2 have been carefully reviewed and incorporated into the discussion section of the manuscript. These revisions address the distribution of samples and the data augmentation process, ensuring that the rationale and methodology are clearly explained.

The sample distribution in Table 1 was designed with careful consideration of the data and study objectives. More samples were allocated to the validation set than the test set to fine-tune hyperparameters during training. This approach improves performance on unseen data and helps prevent overfitting. We also recognize the importance of the test set for obtaining an unbiased estimate of model performance.

During data augmentation, deformations were applied by adjusting the parameter "s" to values greater than zero. Plausible distortions were selected, while higher distortions rendered images unusable, resulting in non-uniform augmentation. This increased the unsupervised training data from 1719 to 4220 (2.76-fold) and the supervised data from 216 to 597 (2.45-fold). Greater augmentation was applied in the unsupervised stage (Barlow twins) to capture feature diversity, while less augmentation was used in the supervised stage to avoid overfitting and ensure generalization.

Comment 5: The authors' response regarding the "Semi-Supervised Model and Architecture" is notably concise and effectively clarifies the methods used in their research, despite being significantly shorter than the corresponding section in the manuscript. I recommend revising the "Architectures" section to mirror this clarity and conciseness, which will enhance the overall readability and precision of the manuscript.

RE: Thank you for your thoughtful recommendation. We have carefully revised the "Architecture" and "Semi-supervised Model" sections, incorporating the suggestions provided to improve clarity and alignment with the reviewer's feedback.

\\subsection*{Semi-supervised model}

We propose a semi-supervised model consisting of two stages. In the first stage, unsupervised learning is used to train various encoders, including ResNet50, ResNet101, DenseNet121, and ResNet101ViT, within a Siamese network architecture based on the Barlow Twins method \\cite{zbontar2021barlow}. This process learns feature representations from a large set of unlabeled images. In the second stage, the encoder weights obtained from the first stage are frozen, and fine-tuning is performed with supervised learning to segment corneal endothelial images using a small number of labeled images.

As shown in Figure \\ref{fig:bd-ssl-models}, the model processes unlabeled images ($X$) through a Siamese network, where each image undergoes geometric transformations (e.g., rotations, flipping, cropping) to generate new samples ($Y^A$ and $Y^B$). These transformed images are passed through identical encoders to produce feature maps ($Z^A$ and $Z^B$), and consistent patterns are identified using the cross-correlation function. The learned weights are then fixed and applied during fine-tuning. A decoder with attention modules and skip connections is added, and a 1×1 convolution at the output generates the final segmentation mask.

\\subsubsection*{Architectures}

We use state-of-the-art pretrained backbones—ResNet50 \\cite{he2015deep}, ResNet101 \\cite{lee2017wideresidualinception}, DenseNet121 \\cite{huang2018densely}, and ResNet101ViT \\cite{chen2022vision}—within our semi-supervised learning framework. These models, trained on large-scale datasets, have proven effective at learning robust features that enhance performance in our corneal endothelial image segmentation task.

ResNet architectures are used for their deep learning capabilities and residual blocks that prevent performance degradation in deep networks. DenseNet is employed for its dense layer connections, allowing the model to capture detailed and contextual features. Finally, ResNet101ViT, which combines ResNet with Vision Transformer (ViT), captures both local and global features by applying attention mechanisms and processing images in patches \\cite{app12188972}. This combination improves feature extraction and segmentation accuracy in medical imaging tasks.

Our approach demonstrates significant improvements in model generalization across different encoders, regardless of the pretrained backbone used.

---

## [Editor Report · Decision Letter 2]

25 Sep 2024

Data augmentation via warping transforms for modeling natural variability in the corneal endothelium enhances semi-supervised segmentation

PONE-D-24-00975R2

Dear Dr. Marrugo,

We’re pleased to inform you that your manuscript has been judged scientifically suitable for publication and will be formally accepted for publication once it meets all outstanding technical requirements.

Kind regards,

Akram Belghith

Academic Editor

PLOS ONE

---

## [Editor Report · Acceptance letter]

24 Oct 2024

PONE-D-24-00975R2 

PLOS ONE

Dear Dr. Marrugo, 

I'm pleased to inform you that your manuscript has been deemed suitable for publication in PLOS ONE. Congratulations! Your manuscript is now being handed over to our production team.

Kind regards, 

on behalf of

Dr. Akram Belghith 

Academic Editor

PLOS ONE